# Characterisation of a soil MINPP phytase with remarkable long-term stability and activity from *Acinetobacter* sp.

**Gregory D. Rix[1], Colleen Sprigg[1], Hayley Whitfield[1], Andrew M. Hemmings**[1,2]**, Jonathan D. Todd[1], Charles A. Brearley**[1] *

**1** School of Biological Sciences, University of East Anglia, Norfolk, Virginia, United States of America,
**2** School of Chemistry, University of East Anglia, Norfolk, Virginia, United States of America

* C.Brearley@uea.ac.uk

**Data Availability Statement:** All relevant data are within the paper and its Supporting Information files.

## Abstract

Phylogenetic analysis, homology modelling and biochemical methods have been employed to characterize a phytase from a Gram-negative soil bacterium. *Acinetobacter* sp. AC1-2 phytase belongs to clade 2 of the histidine (acid) phytases, to the Multiple Inositol Polyphosphate Phosphatase (MINPP) subclass. The enzyme was extraordinarily stable in solution both at room temperature and 4˚C, retaining near 100% activity over 755 days. It showed a broad pH activity profile from 2–8.5 with maxima at 3, 4.5–5 and 6. The enzyme showed Michaelis-Menten kinetics and substrate inhibition ($V_{max}$, $K_m$, and $K_i$, 228 U/mg, 0.65 mM and 2.23 mM, respectively). Homology modelling using the crystal structure of a homologous MINPP from a human gut commensal bacterium indicated the presence of a potentially stabilising polypeptide loop (a U-loop) straddling the active site. By employ of the enantiospecificity of Arabidopsis inositol tris/tetrakisphosphate kinase 1 for inositol pentakisphosphates, we show AC1-2 MINPP to possess D6-phytase activity, which allowed modelling of active site specificity pockets for $InsP_6$ substrate. While phytase gene transcription was unaltered in rich media, it was repressed in minimal media with phytic acid and orthophosphate as phosphate sources. The results of this study reveal AC1-2 MINPP to possess desirable attributes relevant to biotechnological use.

## Introduction

Since the 1990's when the first commercial phytase *Natuphos®* was released to market [1], the market for industrial enzymes has grown to an estimated value of 4.5–5 billion USD (in 2015), of which food and feed applications account for 55–60% [2]. The advent and development of phytases for use in industry is regarded as one of the top ten landmark discoveries in swine nutrition [3]. Alongside benefits to animal nutrition, commercial phytases have also been touted for their environmental benefits, namely the amelioration of phosphate pollution to waterways [4,5].

Phytases in bacteria, fungi, plants and animals [6,7] are commonly classified by protein fold and catalytic mechanism. Four canonical phytases are commonly considered, the β-propeller

**Funding:** GDR was funded by Natural Environment Research Council (NERC) PhD studentships (NERC Doctoral Training Programme grant NE/L002582/1) with support from AB Vista. HW was funded by Natural Environment Research Council (NERC grant: E/W000350/1). The funders had no role in study design, data collection and analysis, decision to publish, or preparation of the manuscript.

**Competing interests:** The authors have declared that no competing interests exist.

**Abbreviations:** BPPhy, β-propeller phytase; BSA, bovine serum albumin; dH2O, deionized water; D/L-Ins(1,2,3,4)P4, 1D/L-*myo*-inositol 1,2,3,4-tetrakisphosphate; D/L-Ins(1,2,4,5)P4, 1D/L-*myo*-inositol 1,2,4,5-tetrakisphosphate; D/L-Ins(1,2,4,6)P4, 1D/L-*myo*-inositol 1,2,4,6-tetrakisphosphate; D/L-Ins(1,2,5,6)P4, 1D/L myo-inositol 1,2,5,6-tetrakisphosphate; D/L-Ins(1,2,3,4,5)P5, InsP5(4/6-OH), 1D/L-*myo*-inositol 1,2,3,4,5-pentakisphosphate; D/L-Ins(1,2,4,5,6)P5, InsP5(1/3-OH), 1D/L-*myo*-inositol 1,2,4,5,6-pentakisphosphate; EDTA, ethylenediamine tetra-acetic acid; GMQE, Global Model Quality Estimate; HAPhy, Histidine Acid Phytase; HCl, hydrochloric acid; HEPES, 4-(2-hydroxyethyl)-1-piperazineethane sulfonic acid; His, histidine; HPLC, high-pressure liquid chromatography; IMAC, immobilized metal affinity chromatography; Ins(1,2,3,4,6)P5, InsP5(5-OH), myo-inositol 1,2,3,4,6-pentakisphosphate; InsP6, phytate, myo-inositol 1,2,3,4,5,6-hexakisphosphate; InsS6, IS6, 1D-*myo*-inositol hexakissulfate; LB, lysogeny broth; MALDI, Matrix Assisted Laser Desorption/Ionization; MINPP, Multiple Inositol Polyphosphate Phosphatase; PAPhy, Purple Acid Phytase; Pi, phosphate; PNPP, *para*-nitrophenyl phosphate; PTPLs, Cysteine phytase Protein Tyrosine Phosphatase like Phytase.

phytases (BPPhy), Purple Acid Phytases (PAPhy), Protein Tyrosine Phosphatase-like Phytase (PTPLPs) (Cysteine phytases) and Histidine Acid Phytases (HAPhys) [8], but the classification has been extended by the characterization of soil metagenomes to include metallo-β-lactamase enzymes [9,10]. The histidine acid phytases are also comprised of a subclass, the Multiple Inositol Polyphosphate Phosphatases (MINPPs), which deviate from HAPhy sequence homology [11]. Although these enzymes were previously considered to function only in animals [12], the first crystallographic study was that of the enzyme from the human gut commensal *Bacteroides thetaiotaomicron*, where the possibility of horizontal gene transfer was mooted [13].

Phytases act through the sequential dephosphorylation of phytate, with different enzymes initiating attack on different positions on the phytate molecule [14]. Many bacterial and fungal phytases are D3-phytases (EC 3.1.3.8), whilst plants as well as *Escherichia coli* possess enzymes commonly called 6-phytases (EC 3.1.3.26). It should be noted however that phytases from plants first described as 1D4-phytases [15] from which the EC 3.1.3.26 designation arises are commonly conflated with 6-phytases (1D6 = 1L4). Enzymes that attack the 5-position have also been identified, including a PTPLP phytase from *Selenomonas ruminantium* [16,17] as well as a lily pollen histidine acid phosphatase of MINPP class [16–19].

The search for more effective phytases with desirable characteristics, improved catalytic activity, heat stability, a wide pH activity profile, enhanced acid- and protease-resistance and cost-effective production has gathered apace [20]. The characterisation of a more effective bacterial phytase from *E. coli* shifted development from fungal sources to bacteria [21], leading to a new generation of enzymes considered superior in several desirable ways such as activity, affinity, proteolytic resistance [5,22].

This search has also been extended into soil environments where *myo-*, *neo-*, *scyllo-* and D-*chiro* forms of inositol phosphates represent substantial reserves of organic phosphate, albeit adsorbed and mostly inaccessible [23]. The soil environment is important for both culture-dependent and culture-independent studies due to the vast diversity of microflora and still untapped potential of soil microbes [24]. These environments have been the target for both metagenomic studies [25,26], as well as culture-dependent phytase isolation studies [27,28]. With phytases from contrasting environments discovered and isolated [29–31], their individual differences can be analysed and used to aid the development and design of more active, thermostable phytases [32,33].

This manuscript details the characterisation of one of the first recombinant MINPP to be isolated from soil *Acinetobacter* sp., that as an exemplar of non-commensal MINPPs has facets of character that illustrate the potential for development of feed enzymes from this sub-class of phytase and for isolation of similar enzymes from soil. The phylogenetic relationship of *Acinetobacter* MINPP phytase to other phytases is shown in Supporting information.

## Materials and methods

### Media

Lysogeny broth was made using sodium chloride, agar (Merck Life Science), tryptone and yeast (Formedium, UK). Na-InsP$_6$ stocks (provided by AB Vista) were prepared in dH$_2$O and adjusted to pH 7 with HCl. Phytic acid sodium salt hydrate (Sigma P8810) was used for the qPCR studies and similarly prepared in dH$_2$O and buffered to pH 7.

### Construction of phylogeny

The phylogeny tree was created using 21 Histidine Acid Phytases (HAPhy), 27 Multiple Inositol Polyphosphate Phosphatases (MINPP), 17 Beta-propeller phytases, 10 Protein Tyrosine Phosphatase-like Phytases and 22 Purple Acid Phytases. The sequences were chosen to

produce a diverse tree both internally and between genes. These were aligned using the online resource MAFFT Version 7 [34] using their automated strategy which choses between, FFT-NS-1, FFT-NS-2, FFT-NS-I or L-INS-I, one of 4 progressive or iterative refinement methods. The Newick output from the alignment was uploaded to the Interactive Tree of Life (iTOL) [35].

## Phytase isolation

*Acinetobacter* sp. AC1-2 was isolated from untilled agricultural soil from Fakenham, UK, purified to single colonies and identified by sequencing of its 16S rRNA gene (GenBank MT450216) and genome (GenBank JABFFO000000000) as described in Rix *et al* [36]. Phytase activity was followed by HPLC [36]. Chromatography data were exported as *x,y* data and redrawn in GraphPad Prism v.6.0 (GraphPad Software, USA).

## Phytase production and purification

The *Acinetobacter* sp. phytase gene (*ac1-2 MINPP*) was identified in the bacterial genome (GenBank JABFFO000000000), with a theoretical protein mass 58.57 kDa, nucleotide and amino acid sequences are presented in the Supporting information. Genomic DNA was extracted as described in the Supporting information. Primers were designed for Gateway cloning in accordance with the Gateway™ Cloning Technology from Life Technologies manual. A two-stage PCR (LR and BP reaction) using the high-fidelity polymerase, Phusion® (NEB), was performed to clone the full gene with the adapters necessary for gateway cloning first into the donor vector pDONR207, followed by cloning into the destination vector pDEST17 (Supporting information). The *ac1-2 MINPP* pDEST17 construct was transformed into Rosetta 2 pLysS (Novagen) for protein expression studies.

The protein purification method is provided in detail in the Supporting information. Briefly, *ac1-2 MINPP* expression was induced in Rosetta 2 pLysS cells. Initial purification efforts expressed a low purity, low activity product. Thereafter the signal peptide was identified using Signal P 5.0 [37], and a new set of primers designed to remove the signal peptide. The protein expressed, hereafter called AC1-2 MINPP was purified using a 1 mL Histrap™ HP column followed by a HilLoad 16/600 Superdex 75 PG column on an ÄKTA pure protein purification system. The purified protein was visualised using SDS-page gel electrophoresis. The protein was sequenced by Protein Mass Fingerprinting using MALDI at the John Innes Centre, Norwich, UK, and compared with the *Acinetobacter* sp. genome to confirm expression.

## Structural biology

Homology modelling of the structure of AC1-2 MINPP was carried out using SWISS-MODEL [38]. Two models of the enzyme less its signal peptide as predicted by SignalP 5.0 [37] were produced based on X-ray crystal structures of the highest scoring sequence homologue, the MINPP from *Bifidobacterium longum* (*Bl*MINPP) with which AC1-2 MINPP shares 34% sequence identity. A first model was based on the structure of apo-*Bl*MINPP (PDB entry 6RXD) [11] and the other on the structure of the enzyme complexed with the non-hydrolysable substrate analogue inhibitor, D-*myo*-inositol hexakissulfate (InsS$_6$) (PDB entry 6RXE). The GMQE scores of the two models were 0.60 and 0.61, respectively. Note that PDB 6RXE shows InsS$_6$ to exhibit static disorder in its complex with *Bl*MINPP. Two conformations of the inhibitor are observed placing the 4- and 6-sulfates in the catalytic A specificity pocket (for an explanation of specificity pocket nomenclature see [11]). Despite this disorder, the sulfate groups in all specificity pockets except D and E are essentially superimposable in the two binding poses. The coordinates of InsS$_6$ were therefore transferred from PDB 6RXE directly to the homology

model of AC1-2 and the sulfate groups substituted with phosphates to generate a model for the AC1-2 complex with phytate. Residues forming the specificity pockets of AC1-2 MINPP were inferred from the predicted structure of its complex with $InsS_6$ and compared with that of *Bifidobacterium longum* (PDB 6RXE), *Bacteroides thetaiotaomicron* (*Bt*MINPP) (sequence identity 21%; PDB 4FDU) [13], and the extracellular histidine phytase from *Aspergillus fumigatus* (sequence identity 15%; PDB 1SK8) [39].

## Phytase assays

Phytase specific activity was determined using the molybdenum blue method for phosphate release [40,41]. One phytase unit is defined as the amount of the enzyme releasing 1 μM inorganic phosphate per minute under the assay conditions. All samples were assayed in triplicate [42].

These reactions were performed from a working stock of AC1-2 MINPP in 25% w/v trehalose unless stated otherwise. In brief, 2.5 μL of 250 nM AC1-2 MINPP was added to 42.5 μL of 0.2 M Na-Acetate pH 5 buffer. The enzyme-buffer solution was mixed with 5 μL of 50 mM $InsP_6$ on ice before being heated at 37˚C for 15 minutes in a PCR machine. Triplicate reactions were performed. The reaction was stopped by addition of 50 μL of a 4:1 ratio of ammonium molybdate sulphuric acid solution, prepared by mixing solutions of molybdate (6 g $NH_4Mo_7O_{24}.4H_2O$ and 22 mL 98% $H_2SO_4$ in 400 mL) with ferrous sulphate solution (2.16 g iron (II) sulphate heptahydrate, 2 drops 98% $H_2SO_4$ in 5mL). Absorbance was measured at 700 nm after 15 minutes and compared against a calibration with $NaH_2PO_4$. AC1-2 MINPP was used at a final concentration of 12.5 nM unless stated otherwise.

## Time-course

For analysis of the inositol phosphate products of AC1-2 MINPP action on $InsP_6$, assays were performed in 0.2 M Na-Acetate pH 5.

## Determination of enantiomerism of principal $InsP_5$ product

The D-and/or L-Ins(1,2,3,4,5)$P_5$ [$InsP_5$ 6/4-OH] product of AC1-2 MINPP action on $InsP_6$ was collected from phytase assays containing 5mM $InsP_6$ substrate in 20 mM Na-Acetate pH 5 buffer incubated for 2h at 31˚C and containing 45–90 nM of glycerol/BSA-stabilized enzyme (0.5 mg/mL BSA, 25% w/v glycerol). The reaction products were resolved by HPLC on a CarboPac PA200 column eluted with HCl [43], but without addition of post-column ferric nitrate. Fractions (1.5 min, 0.6 mL) containing peak(s) were transferred to borosilicate glass tubes, frozen on dry-ice and freeze-dried to remove HCl. The dried samples were rehydrated with 0.5 mL 18.2 Mohm.cm water, aliquots (5 μL) were retained for HPLC, and samples were frozen and freeze-dried again, before rehydration with 50 μL water. An estimated 40 nmol of D-and/or L-Ins(1,2,3,4,5)$P_5$ [$InsP_5$ 6/4-OH] was recovered in a fraction devoid of other inositol phosphates.

The recovered, HPLC-confirmed D-and/or L-Ins(1,2,3,4,5)$P_5$ fraction was presented as substrate to AtITPK1, which we have shown is capable of pyrophosphorylating both $InsP_6$ and D-Ins(1,2,3,4,5)$P_5$ [$InsP_5$ 6-OH] but not D-Ins(1,2,3,5,6)$P_5$ [$InsP_5$ 4-OH] [43]. Briefly, assays of 20 μL volume containing 20mM Hepes, pH 6.5, 1 mM $MgCl_2$, 2 mM ATP, 5 mM phosphocreatine and 3U creatine kinase were supplemented with 5 μM ITPK1 and 0.5 mM substrate. The substrates used were: D-Ins(1,2,3,4,5)$P_5$ as decasodium salt (SiChem GmbH) as a positive control, D-Ins(1,2,3,5,6)$P_5$ as decasodium salt (Sichem GmbH) as a negative control, $InsP_6$, purified from rice bran [44] as positive control and HPLC-purified $InsP_5$ fraction as unknown. Reactions were incubated for 12 h at 25˚C, stopped by addition of an equal volume of 60 mM

$(NH_4)_2HPO_4$, pH 3.5 with orthophosphoric acid, and the whole made up to 70 μL with water. Aliquots (50 μL) were analysed by HPLC on CarboPac PA200 eluted with HCl and subsequent post-column addition of ferric nitrate for detection of inositol phosphates [43].

D-Ins(1,3,4,5)$P_4$ and D-Ins(1,4,5,6)$P_4$, used in other HPLC runs, were obtained from Cayman Chemical, USA, while Ins(1,3,4,6)$P_4$ was obtained from Professor Barry Potter, University of Oxford [43].

## pH profile

The pH profile of AC1-2 MINPP was measured in 0.2M buffer: glycine HCl, pH 2–3.5; sodium acetate, pH 4–5.5; Bis-Tris, pH 6–7; Tris HCl, pH 8–8.5.

## Enzyme activity towards other substrates

Assays were performed as described for phytate with a range of phosphate monoesters.

## Inhibition

The effect of metal ions on enzyme activity was investigated with or without the addition of 1 mM $K^+$, $Mn^+$, $Cu^{2+}$, $Co^{2+}$, $Mg^{2+}$, $Ca^{2+}$, $Zn^{2+}$, $Fe^{2+/3+}$ or $IS_6$, the substrate analogue of $InsP_6$, directly to the reaction mixture.

## Thermostability

The short-term thermostability of AC1-2 MINPP was measured by firstly incubating the enzyme-buffer mixture at 4, 37, 50, 60 and 70˚C for 10 minutes. Following this, activity assays were performed as described above at 37˚C.

## Long-term stability in different storage buffers

The stability of AC1-2 MINPP was measured over an extended period. The protein in gel filtration buffer was mixed in 1:1 ratio with different stabilising agents, 50% (w/v) trehalose, 50% (w/v) trehalose and 1 mg/mL BSA, 50% (w/v) sucrose, 50% (w/v) sucrose and 1 mg/mL BSA, 50% (w/v) glycerol, 50% (w/v) glycerol and 1 mg/mL BSA, 1 mg/mL BSA, or gel filtration buffer, and left at room temperature. The storage protein concentration was 4 μM. On occasions thereafter, aliquots were tested for activity at a final protein concentration of 50 nM.

## Kinetic characterization

AC1-2 MINPP was assayed at twelve substrate concentrations from 12.5–3750 μM $InsP_6$. The progress of reaction curve was fitted to a non-linear regression model for substrate inhibition using GraphPad Prism 8.0.1.

## Measurement of inorganic phosphate content of $InsP_6$ used in gene expression study

The concentration of inorganic phosphate in a 1 mM $InsP_6$ solution (Sigma P8810) was determined, by suppressed ion conductivity HPLC [45], to be 0.365 mM and this concentration of inorganic phosphate was included as a control in the experiments described in Fig 9.

## RNA extraction and quantification

A 10 mL culture of *Acinetobacter* sp. cells growing in either LB or Minimal Media, supplemented or not with 1mM $InsP_6$ or inorganic phosphate (0.365 mM), equivalent to that

impurity in the InsP$_6$, was extracted after reaching late exponential phase. Cells were centrifuged at 6000 x RPM for 5 minutes and to the pellet 1 mL of TRIzol/TRI reagent (Invitrogen) was added. Tubes were vortexed and incubated at room temperature for 5 minutes. Chloroform (0.2mL) was added, the tubes vortexed, incubated at room temperature for 2 minutes and centrifuged at 13,000 x RPM for 10 minutes at 4˚C. Isopropanol (500 μL) was added to the (removed) upper aqueous phase, mixed by inversion and incubated for 10 minutes at room temperature to precipitate RNA. Pelleted (12,000 x RPM, 10 minutes, 4˚C) RNA was washed with 1 mL of 75% ethanol, centrifuged, air-dried, and resuspended in 30 μL RNase-free water.

## DNase treatment

Briefly, 1 μg of RNA, 1 μL DNase (Promega RQ1 RNase-free DNase), 1 μL 10X Reaction buffer were made up to 10 μL with RNAase-free water (NEB) and incubated at 37˚C for 30 minutes, before addition of 1 μL of RQ1 stop solution and further incubation at 65˚C for 10 minutes.

## cDNA synthesis

cDNA synthesis was performed according to the First Strand cDNA Synthesis Standard Protocol from New England BioLabs (NEB) using ProtoScript II Reverse Transcriptase.

Briefly: to 1 μg of DNase-treated RNA, 2 μL Random Hexamers (NEB Random Primer Mix) and 1 μL dNTP mix (10 mM New England Biolabs) was added to a total volume of 10 μL. These were heated at 65˚C for 5 minutes before addition of 4 μL 5X Protoscript II Buffer (NEB), 2 μL 0.1 M DTT, 1 μL Protoscript II Reverse Transcriptase (NEB) and 3 μL RNase-free H$_2$O, to a total volume of 20 μL. Following sequential incubation at 25˚C for 5 minutes and 42˚C for 60 minutes, enzyme was inactivated by heating at 65˚C for 20 minutes. The concentration of cDNA was measured on a NanoDrop (Thermo Scientific) and diluted to a concentration of 250 ng/μL.

## qPCR primer design

qPCR primer sets were designed using the Primer Express Software 3.0.1 with the *ac1-2 MINPP* gene followed by Primer Blast against the *Acinetobacter* sp. genome to discount secondary product formation. The primers used and the conditions of their use are detailed in Supporting information.

## Quantitative PCR

qPCRs were performed for all individual samples in biological and technical triplicates in 20 μL reaction volume, using SensiFast SYBR Hi-Rox Kit (Bioline), with 400 nM primer on a StepOne Plus Real-Time PCR System (Applied Biosystems). The 'housekeeping' gene *RecA* was analysed as reference.

## Data analysis

Statistical analyses were performed using GraphPad Prism 8.0.1. The ΔCt values were first investigated for outliers using the ROUT method, with any outliers being removed from the dataset. The dataset was then analysed for normality and lognormality using the Anderson-Darling test indicating the normalised distribution of the datasets. Significance between the two datasets were analysed using either an unpaired, parametric T-test, or an unpaired non-parametric Mann-Whitney test.

## Results

### Specificity of attack on phytate by Acinetobacter sp. AC1-2 phytase

Confirmation that *Acinetobacter* sp. AC1-2 encodes a MINPP, likely responsible for the ability of the isolate to degrade phytate in solid media and liquid culture was reported [36]. Here we analyse the products of phytate degradation by recombinant AC1-2 MINPP. Fig 1 shows sequential degradation of phytate over a period of 8h with 12.5 nM protein assayed at pH 5. At early stages of degradation three peaks of $InsP_5$ were detected with a predominance of D/L-Ins$(1,2,3,4,5)P_5$ and near equal amounts of D/L-Ins$(1,2,4,5,6)P_5$ and Ins$(1,2,3,4,6)P_5$. The absence of Ins$(1,3,4,5,6)P_5$ among products suggests that the enzyme does not attack the single axial-orientated phosphate on the 2-position. The generation of multiple $InsP_5$s is typical of the commensal bacterial MINPPs characterized to date [11,13,46,47].

Explicit testing of the lack of loss of the 2-phosphate during dephosphorylation has been shown for the related commensal enzyme *Bt*MINPP [13]. Here, we show for AC1-2 MINPP that the $InsP_4$ products of dephosphorylation of $InsP_6$ do not co-elute with Ins$(1,3,4,5)P_4$ or Ins$(1,4,5,6)P_4$, or their enantiomers which are unresolvable (Supporting information). One of the $InsP_4$ products co-eluted with Ins$(1,3,4,6)P_4$ which is a *meso*-compound (Supporting information) but as this compound co-elutes with Ins$(1,2,3,4)P_4$ and its enantiomer Ins$(1,2,3,6)P_4$ (Supporting information) we cannot assign identity unequivocally. Of the fifteen stereoisomers of $InsP_4$, only Ins$(1,3,4,5)P_4$, Ins$(1,4,5,6)P_4$, or their enantiomers, and Ins$(1,3,4,6)P_4$ possess a 2-phosphate. Nevertheless, the absence of detectable Ins$(1,3,4,5,6)P_5$ in more than thirty independent AC1-2 MINPP assays with different degrees of dephosphorylation leads us to conclude that AC1-2 MINPP lacks $InsP_6$ 2-phosphohydrolase (phytase) activity.

The co-production of $InsP_4$ with $InsP_5$ at 30 min, before the peak of accumulation of $InsP_5$, suggests that, even in the presence of excess $InsP_6$, $InsP_5$s are better substrates than $InsP_6$. Multiple peaks of $InsP_4$ co-exist in the degradation products with $InsP_5$ and $InsP_3$ products, until the $InsP_5$s are wholly consumed at which point a single peak of D/L-Ins$(1,2,4,6)P_4$ was observed. $InsP_4$s are particularly well resolved on the CarboPac PA200 column [44] allowing identification of the major and minor routes of initial and subsequent dephosphorylations. $InsP_3$, of which there are 20 possible stereoisomers, is less well resolved on this column, but it is evident that as $InsP_4$ degradation proceeds to exhaustion–one of the two resolvable peaks (which could contain multiple isomers) predominates, concurrent with the appearance of a single peak of $InsP_2$. While there are fifteen possible stereoisomers of $InsP_2$, we may reasonably assume that the peak is comprised of species retaining the 2-phosphate. Similarly, we may assume that the monophosphate product is Ins2P, which along with other monophosphates co-elutes with Pi at the solvent front of this HPLC system–accounting for the progressive accumulation of Pi over the course of the assay.

The lack of HPLC matrices for separation of inositol phosphate enantiomers precludes us from simple chromatographic determination of the ability, or otherwise, of the enzyme to discriminate between enantiotopic D4- and D6-positions or D1- and D3-positions. We, therefore, sought alternative approach to the problem.

### Application of a stereospecific pyrophosphorylating enzyme activity to identification of $InsP_5$ products of AC1-2 MINPP action

We have shown that AtITPK1 pyrophosphorylates $InsP_6$ and D-Ins$(1,2,3,4,5)P_5$ [$InsP_5$ 6-OH] exclusively among $InsP_5$s [43]. We exploited this activity for the characterization of this member of the MINPP class. The D-and/or L-Ins$(1,2,3,4,5)P_5$ [$InsP_5$ 6/4-OH] product of AC1-2 MINPP action on $InsP_6$ was purified by HPLC, desalted and the recovered fraction shown to

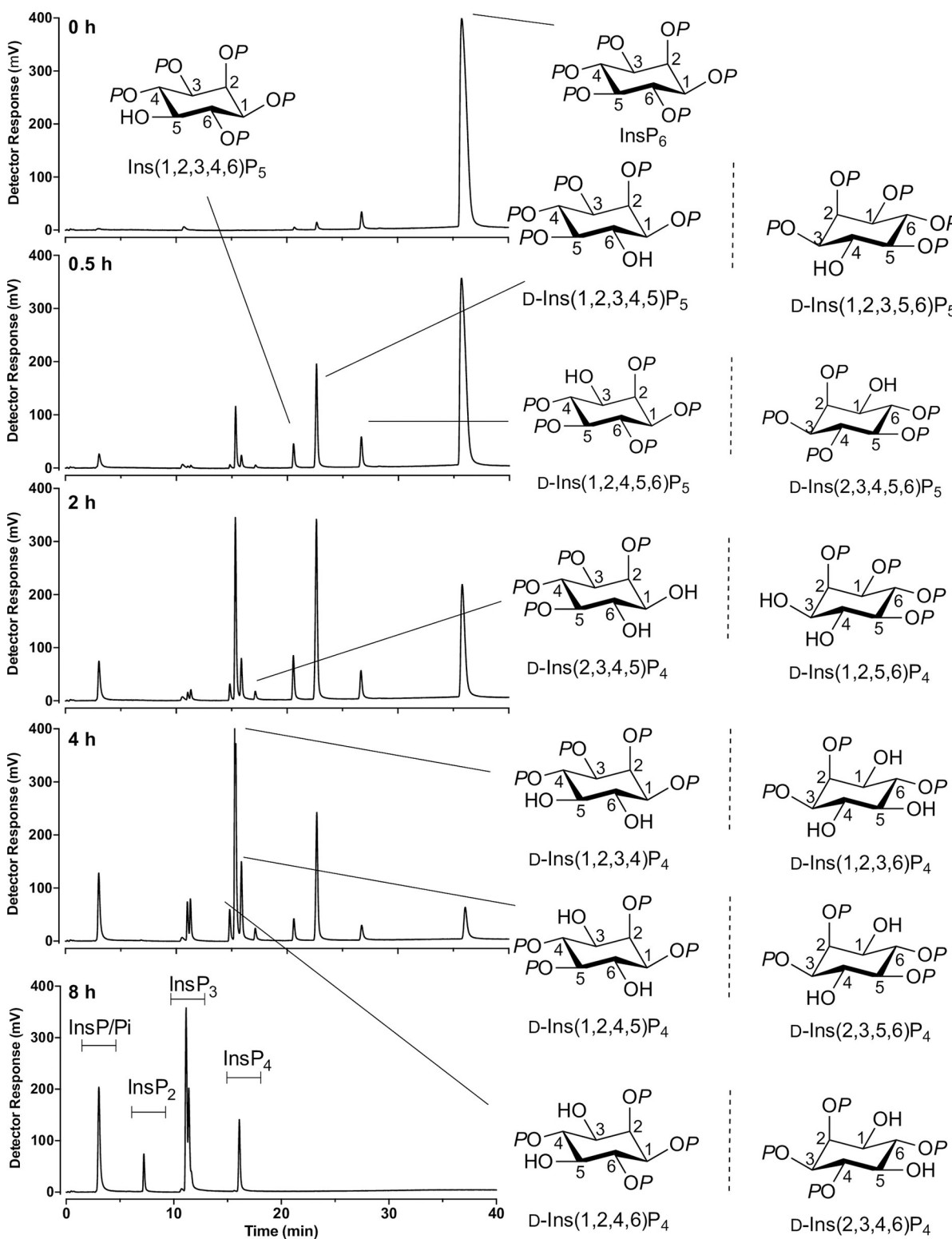

**Fig 1. Degradation of phytate by AC1-2 MINPP.** Degradation was followed by HPLC at 30 min, 2h, 4h and 8h. Structures of inositol phosphate products (as pairs of enantiomers where appropriate, named by D-notation) are shown.

be devoid of other inositol phosphates (Supporting information). The pyrophosphorylating activity of ITPK1 was tested with InsP$_6$, D-Ins(1,2,3,4,5)P$_5$ [InsP$_5$ 6-OH] and D-Ins(1,2,3,5,6) P$_5$ [InsP$_5$ 4-OH], beside the recovered InsP$_5$ fraction. InsP$_6$ was converted in 90% yield to 5-InsP$_7$ (5PP-InsP$_5$, see [43]] (Fig 2C and 2D), D-Ins(1,2,3,4,5)P$_5$ was converted to pyrophosphate product (PP-InsP$_4$) in 58% yield (Fig 2G and 2H), InsP$_6$ is a better substrate, while D-Ins(1,2,3,5,6)P$_5$ was not modified by ITPK1 (Fig 2E and 2F). The recovered InsP$_5$ fraction was converted in approximately 12% yield to a pyrophosphate product (a PP-InsP$_4$) that eluted (on CarboPac PA200) mid-way between Ins(1,3,4,5,6)P$_5$ [InsP$_5$ 2-OH], the last-eluting InsP$_5$, and InsP$_6$ with identical retention time to the PP-InsP$_4$ generated from D-Ins(1,2,3,4,5)P$_5$ (Fig 2A and 2B) (after [43]). These experiments confirm that AC1-MINPP has D6-phytase activity and rationalize modelling of InsP$_6$ substrate interaction with AC1-MINPP (see below). The lack of pyrophosphorylating enzymes with enantiomeric preference for D-Ins(1,2,3,5,6)P$_5$ (over D-Ins(1,2,3,4,5)P$_5$) precludes an equivalent approach that could unequivocally identify D4-phytase activity for AC1-2 MINPP, though we cannot discount it.

Structures of *Bl*MINPP-IS$_6$ complexes (PDB 6XRE [11]) show static disorder with ligand bound in two orientations in different monomers of the crystallographic asymmetric unit. The similar occupancy ratios of ligands placing the sulfate equivalent of 1D-4 phosphate and 1D-6 phosphate (of InsP$_6$) close to the catalytic histidine in the different units may indicate that the enzyme has little discrimination between attack on the 1D4- and 1D6-positions [11]. Considering these observations, the major and minor routes of the first two dephosphorylations catalysed by AC1-2 MINPP are summarized in Fig 2.

## Structural features of AC1-2 MINPP: Homology modelling predicts a polypeptide insert in AC1-2 MINPP that spans the active site

The closest sequence homologue to *AC1-2 MINPP* for which a high-resolution structure exists in the PDB was found to be the MINPP from *Bifidobacterium longum*, *Bl*MINPP, (Supporting information), the proteins share 34% sequence identity. Homology modelling was used to generate models for the structure of AC1-2 MINPP in the apo-state and as bound to InsS$_6$. The overall structure of AC1-2 MINPP is therefore predicted to resemble *Bl*MINPP having α/β- and α-domains with an active site arranged between the two domains.

Phylogenetic analysis has revealed three groups of polypeptide inserts in MINPP sequences named U-loops [11] which have been given the identifiers A, B, or C depending on insert length. Possessing a lipoprotein-like SEC/SPII signal peptide, AC1-2 MINPP has a 41-residue polypeptide insertion in the α-domain that maps to the type A-type U-loop found in the MINPP from *Bifidobacterium longum* (*Bl*MINPP) (Fig 3A). The insertion is followed immediately by a characteristic tetrapeptide motif (DAAM in *Bl*MINPP and DAAA in AC1-2 MINPP), which is absent in sequences that do not contain a U-loop. The AC1-2 MINPP insertion is, however, shorter than the type A-type U-loop in *Bl*MINPP by eight residues and lacks cysteine residues that form a disulphide bridge in the latter. The U-loop residues in *Bl*MINPP span the active site and close down onto a modelled bound InsS$_6$ substrate analogue (and by inference onto a bound substrate molecule) through a rigid body motion involving a major part of the α-domain [11]. The prediction of a large type A U-loop in AC1-2 MINPP strongly suggests the presence of similar rigid body domain motions, presumably to allow the imposition of additional contacts with phytate in the complex, particularly in specificity pocket D (Fig 3B). However, it is relevant to note that the nature of the homology modelling process, particularly with respect to the prediction of the conformation of large polypeptide loops, leaves room for considerable uncertainty in the conformation of the U-loop in AC1-2.

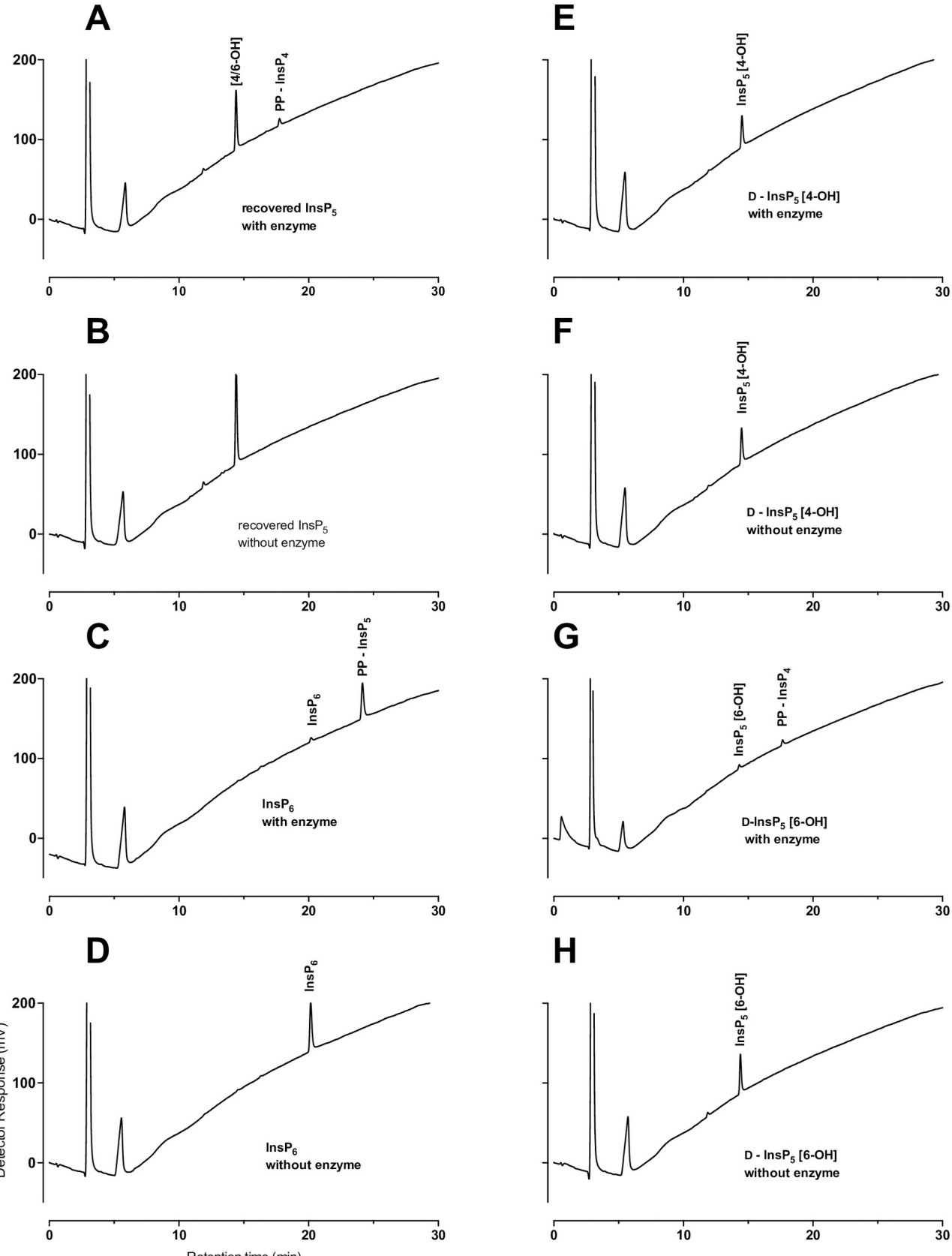

**Fig 2. Schematic detailing potential pathways of degradation of phytate by AC1-2 MINPP.** Predominant products are indicated by size of font; * indicates the predominant InsP$_5$ impurity in the substrate; weight of arrow indicates likelihood of route of degradation.

## Specificity pocket content provides insights to the residue determinants of AC1-2 MINPP positional specificity

Phytases can be grouped according to the specific position of the phosphate ester group on the phytate molecule at which hydrolysis first occurs. Accordingly, phytases which are principally involved in phytate mineralization show high stereospecificity and can be described as 3-phytases (EC 3.1.3.8) or 6-phytases (EC 3.1.3.26). MINPPs, on the other hand, show lower stereospecificity and hydrolyze phytate to generate a mixture of inositol pentakisphosphates. AC1-2 behaves as a typical MINPP and displays 6-phytase activity (Fig 2). We employed homology modelling and structural alignment in an effort to investigate the residue determinants of AC1-2 MINPP positional specificity, employing the specificity pocket nomenclature adopted by Acquistapace *et al* [11] in study of the cell-surface anchored MINPP from *Bifidobacterium*

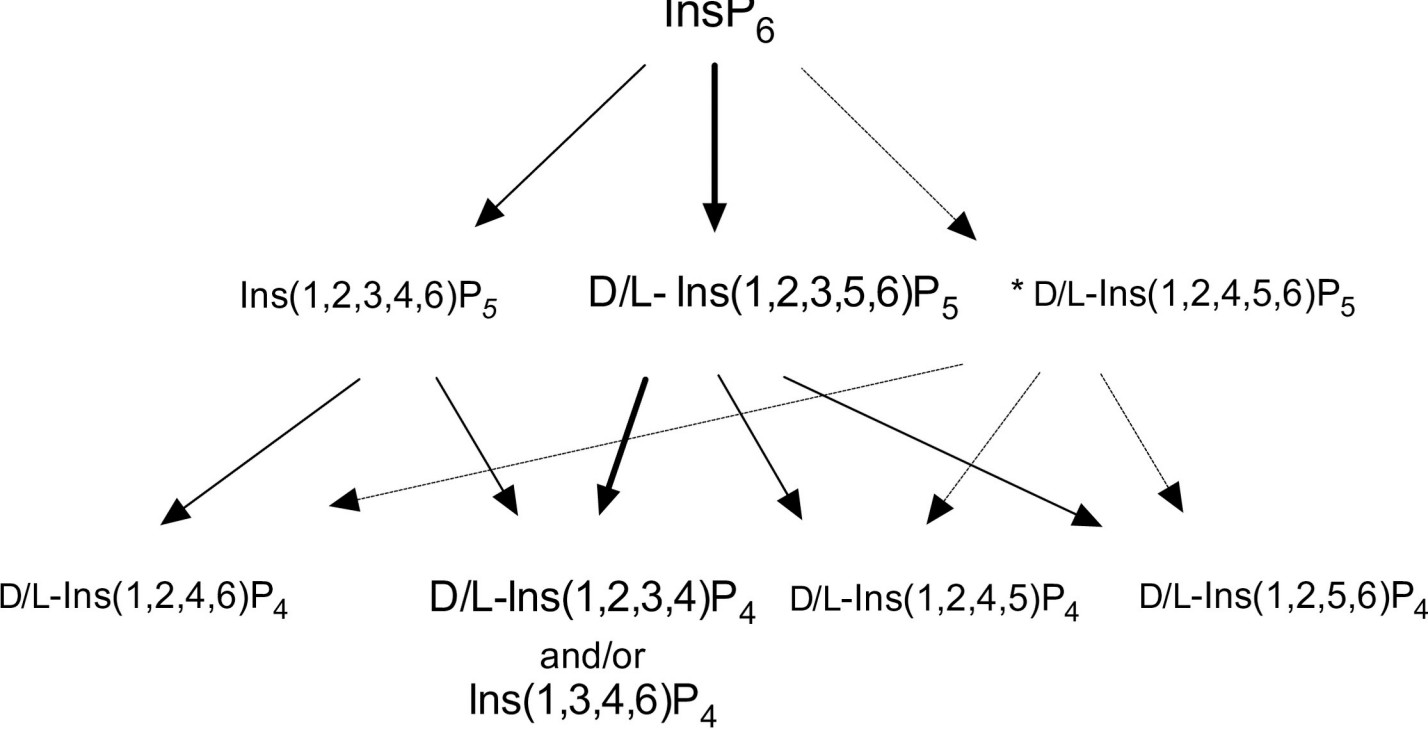

**Fig 3. Predicted structural features of AC1-2 MINPP: U-loop and specificity pockets. A,** Alignment of the amino acid sequences of AC1-2 MINPP and *Bl*MINPP in the region of the U-loop. A blue box delimits U-loop residues. Cysteine residues forming a disulphide bridge in the crystal structure of *Bl*MINPP (PDB 6XRE) are highlighted yellow. Conserved residues (in red) are part of a MINPP-specific tetrapeptide motif. Positions of residues contributing to specificity pockets are indicated by inverted blue triangles. **B,** Molecular surface representations of the structures of apo- (left) and InsS$_6$-bound (right) AC1-2 MINPP predicted by homology modelling. The U-loop residues are coloured green with the remainder of the molecule in cyan. Atoms of the substrate analogue inhibitor, InsS$_6$, are shown as spheres and coloured red (oxygen), cyan (carbon) and orange (sulphur). **C,** Residues predicted to contribute to the specificity pockets of AC1-2 MINPP and selected histidine phytases. AC1-2, *Bl* and *Bt* are the MINPPs from *Acinetobacter sp.* AC1-2 (this study), *Bifidobacterium longum* (PDB 6XRE) and *Bacteroides thetaiotaomicron* (PDB 4FDU), respectively. *Af* is the histidine phytase from *Aspergillus fumigatus* (PDB 1SK8). Specificity pockets are labelled A-F as described by Acquistapace *et al.* (2000). Each alignment shows spatially equivalent residues in the specificity pockets of each enzyme, which lie within 5Å of the phosphorus of the corresponding phosphate group on the substrate (the positions of sulphate groups of the inhibitors in each structure are taken to be the same as the phosphate groups of phytate. Numbering is according to the AC1-2 sequence. Residues that are completely conserved are highlighted in bold with red text. Red arrows indicate residues that have the closest interactions with the substrate analogue inhibitor. Note that residues 298 and 301 contributing to the D-pocket are found on the U-loop. Residues marked with an asterisk in pockets D and E are predicted to not interact with the substrate when bound with the 4-phosphate in the A-pocket.

*longum* [11]. Alignment of the structures of sequence homologues to that of AC1-2 MINPP revealed residues forming the specificity pockets in related clade 2 histidine phytases [48] (Fig 3A). The enzymes chosen for analysis were homologues to AC1-2 MINPP for which high-resolution crystal structures of their complexes with $InsS_6$ were available. These include the MINPPs from *Bifidobacterium longum*, a predominant 4/6-phytase [11], and *Bacteroides thetaiotaomicron*, a predominant 5-phytase [13]. We also included the stereospecific histidine phytase from *Aspergillus fumigatus* [39]. This D3-phytase (EC: 3.1.3.8) is structurally more similar to MINPPs, as a class, than is *the E. coli* D6-phytase [11].

Consideration of the spatially conserved residues in the different specificity pockets gives possible insights into the molecular basis for the D6-phytase activity of AC1-2 MINPP. Note that *myo*-inositol hexakissulfate ($InsS_6$) is reported to act as a competitive inhibitor of phytases and in crystal structures is assumed to mimic the substrate by adopting a pseudo-productive binding [11,13,49,50]. In the following discussion, the positions of the sulfate groups of $InsS_6$ were therefore taken to represent the phosphate groups of phytate as bound to the enzymes. The results of the following analysis must therefore be taken as indicative.

The scissile phosphate group of the substrate occupies pocket A (Fig 3C). The phosphate here is intimately bound, making six polar contacts with pocket residues. The role of the pocket is to engage with the scissile phosphate and to position and orient it for hydrolysis, consequently, pocket A residues are highly conserved between the enzymes considered. Residues in pockets B, C and F have fewer contacts with their corresponding phosphate groups but those residues involved in direct polar contacts with the substrate analogue (and therefore by inference with the substrate), while predominantly conserved with *Bl*MINPP, vary in the *B.thetaiotaomicron* and *A.fumigatus* enzymes. Given the observed D6-phytase positional specificity of AC1-2 MINPP (Fig 2), interactions at residues 18, 199 and 352 (AC1-2 numbering) can therefore be identified as candidates for residue determinants of positional specificity in this subset of enzymes. Indeed, the mutation R183D in *Bt*MINPP (equivalent to residue 199 in AC1-2 MINPP) converts *Bt*MINPP, a predominant 5-phytase, to an *A.fumigatus*- or Klebsiella-like D1/3-phytase, we use the term 'D1/3-' to indicate the unknown enantiospecificity of *Bt*MINPP, in contrast to the known D3-specificity of the Aspergillus and Klebsiella enzymes [39,51].

## Biochemical properties of AC1-2 MINPP

Initial purification efforts yielded an impure protein with low specific activity (13 U/mg). Therefore, purification was repeated after expression from a construct lacking signal peptide. The profile of $InsP_6$ degradation products produced by AC1-2 MINPP mirrors that of *Acinetobacter* sp. culture incubated with $InsP_6$ (*cf.* Fig 1 and Rix *et al* [36], Fig 4).

The ability of AC1-2 MINPP to use other commonly assayed substrates, adenosine triphosphate, glycerol 3-phosphate, glucose 6-phosphate, pyrophosphate, *para*-nitrophenyl phosphate and creatine phosphate was tested (Fig 4A). Within the HAPhy classification there are two subgroups, those that have a broad substrate specificity but a low specific activity and conversely, those that have a narrow substrate specificity and a high specific activity towards phytate [48]. AC1-2 MINPP is in the latter group, exhibiting narrow substrate specificity with relatively low activity towards other phosphorylated substrates such as *para*-nitrophenyl phosphate, 25.8%; and glycerol 3-phosphate 5.4%; in relation to $InsP_6$. This specificity is in agreement with previous analyses of wheat, barley and avian MINPPs [12,52].

Prior to the post-genomic era, characterization of enzymes including phytases commonly tested the effect of metal ions and substrate analogues or residue-modifying reagents to probe reaction mechanisms. Such experiments have a contemporary relevance in consideration of

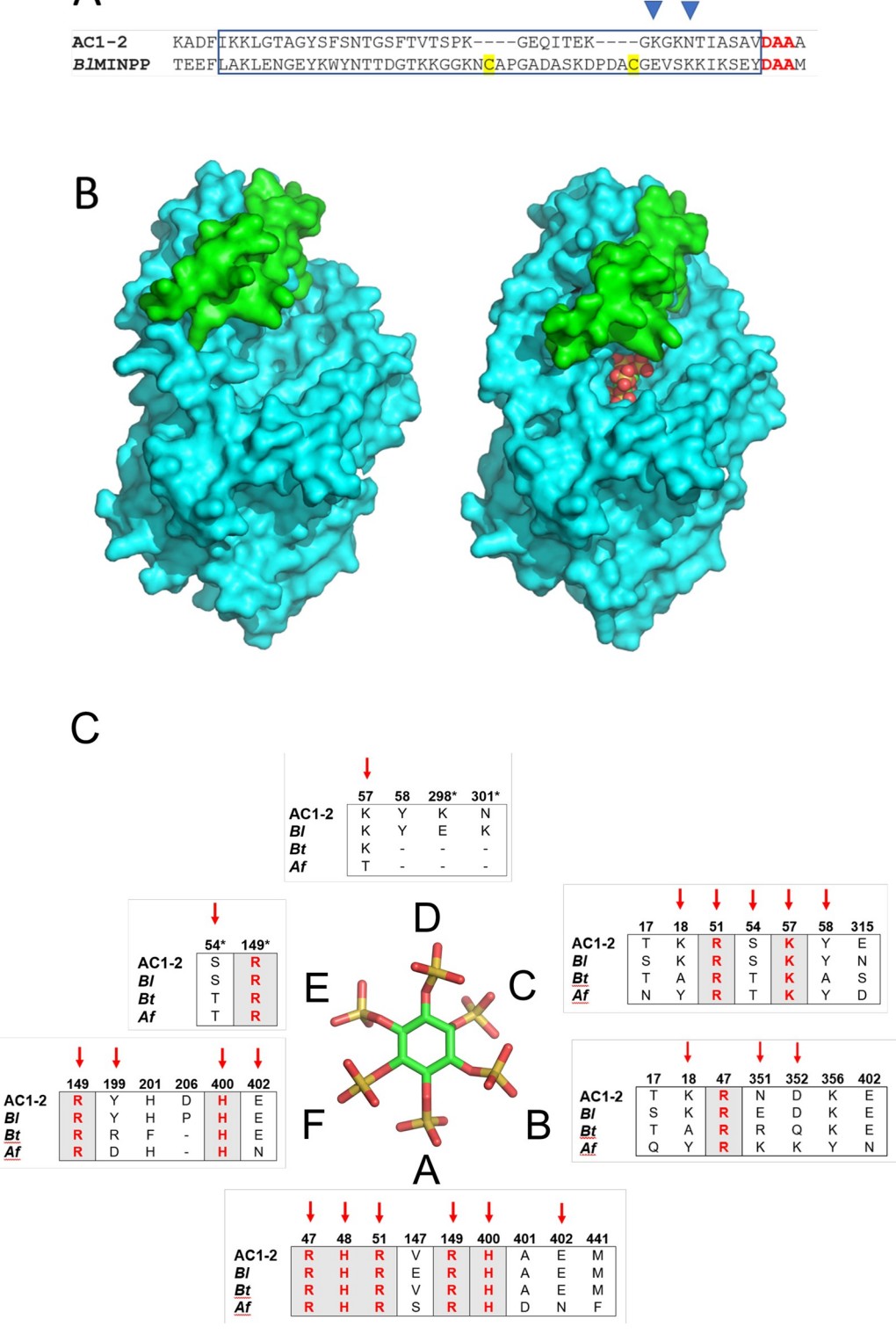

**Fig 4. Biochemical characterisation of AC1-2 MINPP. A,** The specificity of the enzyme towards different phosphate (P)—containing compounds assayed at 1mM. Significant differences between compound and InsP$_6$ are indicated at P≤0.05, *; P≤0.01, **; P≤0.001, *** and P≤0.0001, ****. **B,** inhibition of activity by metal ions. Significant differences between metal ions and control are indicated at P≤0.05, *; P≤0.01, **; P≤0.001, *** and P≤0.0001, ****. **C,** pH-activity profile; **D,** thermostability. A-D, means and standard deviation of three measurements.

biotechnological end use of phytases, in production systems and use in animals. We therefore tested potential metal ion inhibitors and substrate analogue at a concentration of 1 mM for effect on AC1-2 MINPP activity (Fig 4B). Metal ions are known to affect the activity of phytases, these may be due to the inability of the phytase to act upon metal ion-phytate complexes [53]. InsS$_6$ is commonly used in crystallographic studies to aid in the identification of key active site residues [54]. Fe$^{2+/3}$ showed the strongest inhibition of enzyme activity, $29 \pm 6\%$ by 1 mM followed by Cu$^{2+}$ > Mn$^+$ > Co$^{2+}$ > InS$_6$ > Mg$^{2+}$ > K$^+$ > Zn$^{2+}$ > nil addition and Ca$^{2+}$.

The pH profile of candidate feed enzymes is an important parameter. AC1-2 MINPP showed a broad pH-activity profile with three maxima at pH 3, 4.5–5 and 6 (Fig 4C).

To determine the thermostability of AC1-2 MINPP, protein was incubated at temperatures in the range 4–70°C before the addition of InsP$_6$ to assay activity (Fig 4D). Activity was abolished after incubation at 60°C and was reduced to $40 \pm 3\%$ to that of the control sample at 50°C.

## Long term stability of AC1-2 MINPP

A long-term stability experiment was performed on AC1-MINPP. Aliquots of the protein preparation were stored at room temperature in a range of stabilising solutions or buffer only (control) and activity measured at intervals. AC1-2 MINPP showed a remarkable resilience to periodic fluctuations in room temperature during summer and winter months, with the protein retaining activity during two summer "heatwaves" where average room temperatures in the lab moved over 30°C (Fig 5A). We have no obvious explanation of the 'dip' in activity at 220 days and can only speculate that over the course of the observations differently calibrated pipettes might have been used.

A similar experiment with a full-length, signal peptide-containing, and somewhat impure protein preparation showed low activity, around 13 U/mg at 37°C, that was maintained at room temperature and at 4°C (Supporting information). This protein yields products (Supporting information) that are identical to those obtained from AC1-2 MINPP (Fig 6). The experiments to determine enantiospecificity shown in Fig 6 were performed on glycerol/BSA-stabilized enzyme (lacking signal peptide) stored on the bench for > 3 years. Denaturing (SDS) polyacrylamide gel electrophoresis of the signal peptide-lacking protein, at point of preparation, is shown (Fig 5A). A similar analysis of the protein yielding results of Fig 6 is shown (Fig 5C). The greater than 3 year stability/activity of this enzyme is remarkable.

AC1-2 MINPP was also assayed for analysis of kinetic parameters. The enzyme shows Michaelis-Menten kinetics and substrate inhibition, with a specific activity, V$_{max}$, of 228 U/mg, K$_m$ of 0.65 mM and K$_i$ 2.23 mM, respectively (Fig 7).

## Regulation of ac1-2 MINPP by substrate and product

In the absence of literature reports of the control of expression of bacterial MINPPs, we designed primer sets to determine the response of *ac1-2 MINPP* to growth condition. We chose rich and minimal media with or without the presence of InsP$_6$, comparing *ac1-2 MINPP* expression to the 'housekeeping' gene *RecA*. cDNA from *Acinetobacter* sp. AC1-2 grown to exponential/late exponential was analysed for changes in expression. Due to the impurities commonly found in commercial bought InsP$_6$, the concentration of inorganic phosphate therein was measured and equivalent concentration supplemented into minimal media [55]. *ac1-2 MINPP* expression was unaltered by InsP$_6$, and associated phosphate, in LB. In minimal media, the inclusion of inorganic phosphate or InsP$_6$ (containing inorganic phosphate impurity) reduced gene expression ($p < 0.05$) by 2.6-fold and 7.5-fold, respectively (Fig 8).

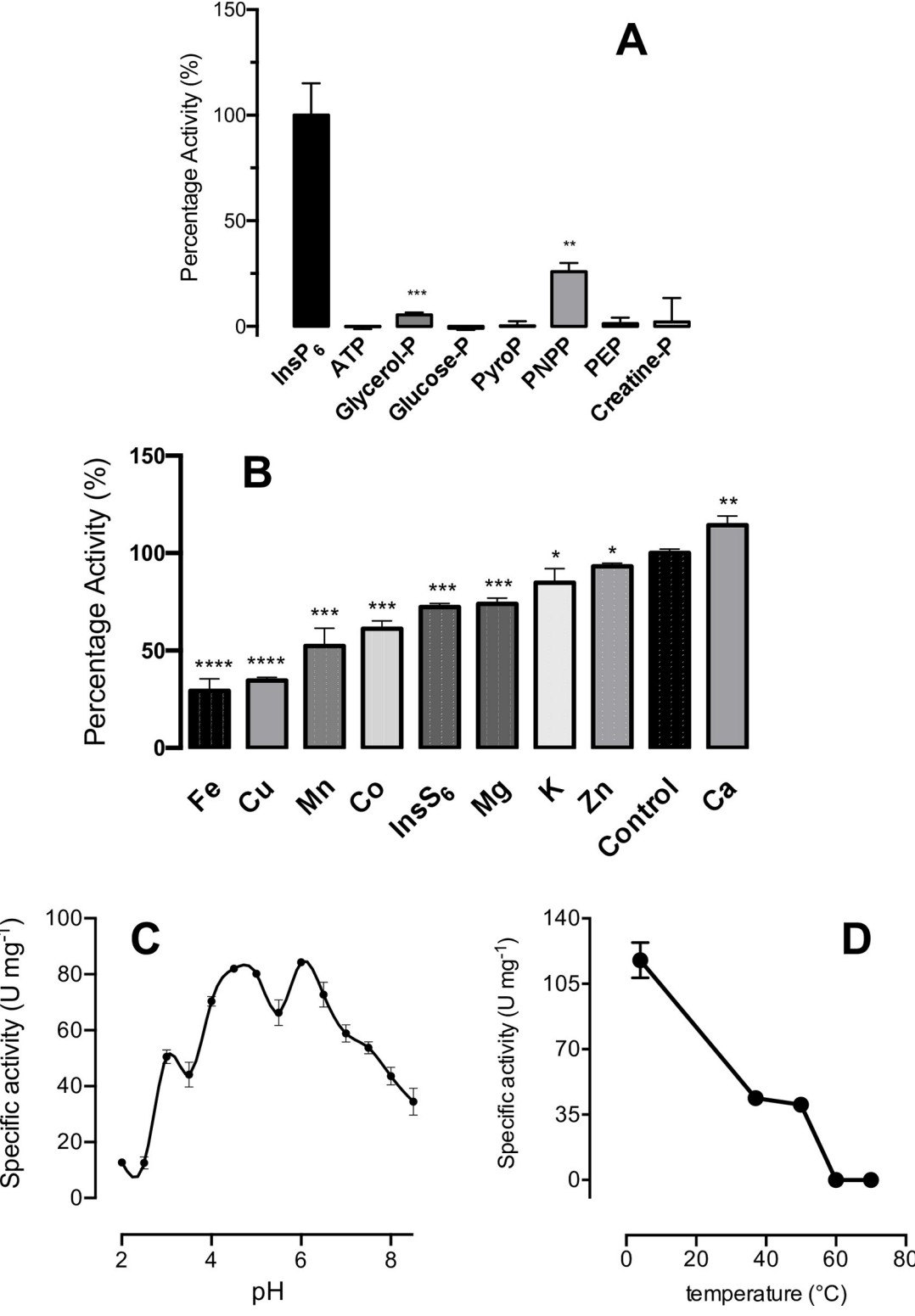

**Fig 5. Long-term stability of AC1-2 MINPP.** Stability was assessed in seven different stabilising solutions and a control (gel filtration buffer) during storage in ambient conditions. **A,** Isolation of AC1-2 MINPP through a two-step, Ni-affinity and size exclusion, purification. Aliquots of protein taken at different stages of the purification were subjected to SDS-PAGE on a 12% gel. Lanes labelled 1–9 are flanked, left, by molecular mass markers identified by mass (kDa). Lanes: 1 and 2, uninduced Rosetta ™ 2 (pLysS) cells; 3 and 4, 0.1 mM IPTG-induced cells; 5, crude lysate of concentrated, induced cells; 6, cell pellet; 7, clarified cell

lysate (supernatant); 8, Amicon® Ultra-15 (10 kDa cut-off) -concentrated fraction, post Ni-affinity chromatography; 9, Amicon-concentrated fraction, post size exclusion chromatography. **B**, Enzyme activity of AC1-2 MINPP during prolonged storage at 4˚C or ambient conditions (occasionally reaching 30–35˚C). Protein was stored at a concentration of 4 μM in 50 mM Tris-HCl pH 7.5 300 mM NaCl (control) or in 25 mM Tris-HCl pH 7.5 150 mM NaCl with stabilising agent as indicated. Error bars show standard deviation of triplicate measurements. **C**. SDS-PAGE of AC1-2 MINPP stored for greater than three years at a concentration of 4 μM in 25 mM Tris-HCl pH 7.5 150 mM NaCl with stabilizing agents as indicated. Thereafter, aliquots of protein were subjected to SDS-PAGE on a 12% gel, Lanes 1–4, flanked on the left by molecular mass markers (kDa). Lanes, stabilising agents: 1, 30% w/v glycerol; 2, 30% w/v glycerol and 0.5 mg/ml BSA; 3, 30% w/v sucrose; 4, 30% w/v sucrose and 0.5 mg/ml BSA.

## Discussion

AC1-2 MINPP is one of the first phytases of this class isolated from the soil environment. MINPPs are related to branch 2 of the histidine acid superfamily, which contains the histidine acid phosphatases and phytases [48]. HAPhytases are characterised by their active site hepta-peptide sequence motif RHGxRxP and proton donor motif HD, and a large α/β domain and a α domain. The AC1-2 phytase contains a slightly different heptapeptide sequence motif, RHGSRGL, and has a tripeptide protein donor motif of HAE. Additionally, it contains several MINPP-specific motifs that are not found in HAPhys: PMAAN and LYNE are located on the β-sheets of the α/β domain; the methionine residue of PMAAN forms part of substrate speci-ficity pocket A. The majority of MINPP studies describe a protein that is commonly found in eukaryotic organisms [12,52] but which has also been described in gut commensal bacteria [13,46].

It may be these unique properties which allow the soil derived MINPP to function over a wide range of pH, similar to the pH range in the digestive tract [56]. It is unusual that AC1-2 MINPP displays multiple maxima, with many bacterial phytases usually exhibiting either one or two maxima [57,58]. This phytase showed highest activity at pH 6 which deviates slightly from the optimal pH of histidine acid phosphatases which is typically within the pH range of 2.5–5.5, however this is not uncommon for many isolated phytases which have their pH optima in the range of 4.5–6. Indeed, it may be a feature of this relatively under-examined MINPP class.

Phytases showing a diverse range of pH activity are preferable in animal feed applications as they must remain optimally active in the digestive tract [59]. AC1-2 MINPP shows contin-ued activity for over 755 days in purified form at room temperature, in some cases, activity did not decrease. Bovine serum albumin as a co-protectant proved to be an excellent stabilising molecule maintaining higher enzyme activity in all treatments to which it was added.

The inhibition of AC1-2 MINPP was tested using a range of metal ions and the substrate analogue InS$_6$. Fe$^{2+/3+}$ and Cu$^{2+}$ reduced activity to $<$ 50%, at 1mM, whereas Zn$^{2+}$ ion was without strong effect. Typically, zinc is referenced as one of the most potent phytase inhibitors [52,60], however in this instance, activity was only reduced to 93%. Furthermore the substrate analogue IS$_6$, also regarded as a potent inhibitor [61], reduced activity to only 72%. Activity was enhanced slightly, to 114% by calcium, a facet of character ordinarily associated with Beta-propeller phytases that have structural calcium ions [62]. Calcium is a critical component of feed matrices employed in poultry feed trials, with calcium, phytase and available phosphate carefully titrated to achieve optimum degradation of phytate in feed [63]. Laying hen diets typ-ically include $>$ 10% limestone [63]. AC1-2 MINPP also showed a relatively narrow substrate specificity, favouring InsP$_6$ with minimal degradation of other glycerol 3-phosphate.

AC1-2 MINPP shows substrate inhibition (K$_i$ 2.2 mM). Substrate inhibition by phytate has been well documented [56]. The fungi *A. ficuum* and bacteria *C. braakii* YH-15 phytases showed inhibition at phytate concentrations above 1.2 mM and 1.5 mM respectively [64,65].

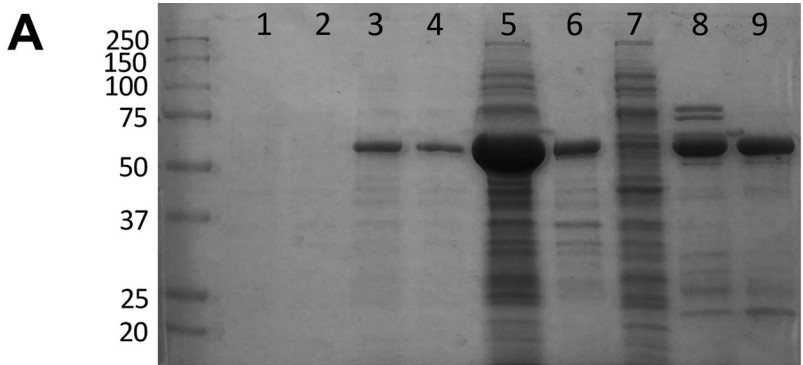

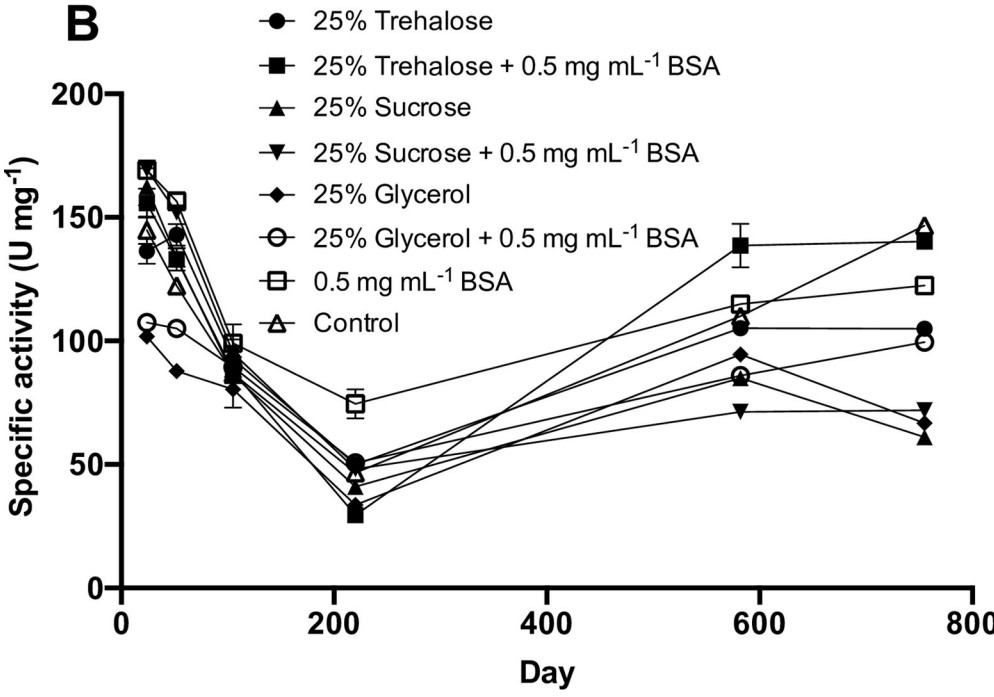

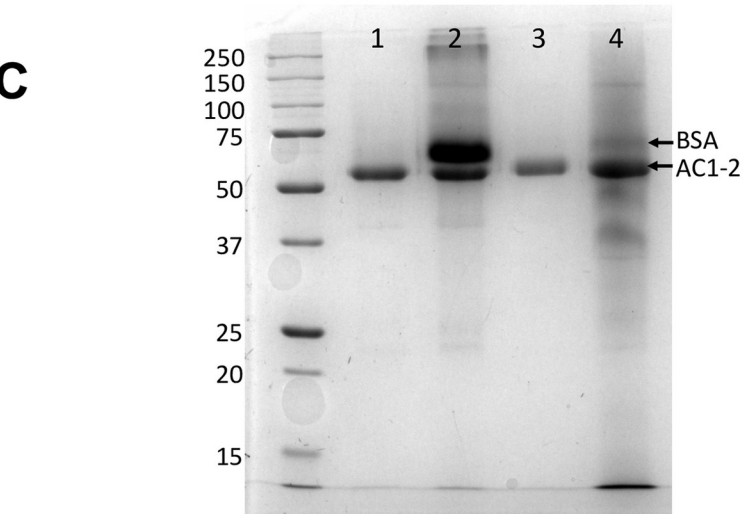

**Fig 6. Enantiospecificity of AC1-2 MINPP attack on phytate.** HPLC of: **A**, products of incubation of a purified and desalted D-and/or L-Ins(1,2,3,4,5)P$_5$ [InsP$_5$ 4/6-OH] fraction generated by AC1-2 MINPP with AtITPK1; **B**, a no-enzyme control for A; **C**, products of incubation of InsP$_6$ with AtITPK1; **D**, a no-enzyme control for C. **E**, products of incubation of D-Ins(1,2,3,5,6)P$_5$ (InsP$_5$ [4-OH]) with AtITPK1. **F**, a no-enzyme control for E. **G**, Products of incubation of D-Ins(1,2,3,4,5)P$_5$ (InsP$_5$ [6-OH]) with AtITPK1. **H**, a no-enzyme control for G. Approximately, one third of sample equivalent (to A-E, and H) was injected for G. The units and scales of panels (A-H) are identical.

The K$_m$ (0.65 mM) of AC1-2 MINPP is typical of the values reported (0.15–1.37 mM) for a range of anonymous commercial phytases [66]. The V$_{max}$ value of 228 U/mg, while lower than that (1123 U/mg) of the codon-optimized histidine acid (HD) phytase AppaAs-OP (optimized for expression in *Komagataella phaffii* [66] is greater than that [23–196 U/mg] of a range of fungal phytases [59,67] with which AC1-2 MINPP and other bacterial MINPPs [11] show greater structural similarity, but again less than that ($\sim$2,000 U/mg) of *Peniophora lycii* [68]. The extreme stability of this enzyme, greater than 3 years on the bench, show that the MINPP scaffold is a very good starting point for engineering this class of enzyme, or for searches for related enzymes. The commercial choice of enzyme rests on much more than simple specific activity. Choice is tempered by expression host, cost effective processing and opportunity for engineering thermostability and protease resistance. Indeed, current commercial enzymes have been engineered to increase thermostability. In the case of *E coli* AppA-derivatives T$_m$ has been increased by more than 12 degrees [69].

In bacteria, phytase is the product of an inducible gene that may be subject to complex regulation [70,71]. A general feature commonly observed in microbial phytase producers is the regulatory inhibition of phytase production by inorganic phosphate levels [71]. This may have an effect on the RT-qPCR results displayed (Fig 8) as many of the commercial phytates available contain high levels of inorganic phosphate or other inositol phosphates [44,72]. Nevertheless, our data show dual aspect to regulation of AC1-2 MINPP phytase activity, by inorganic

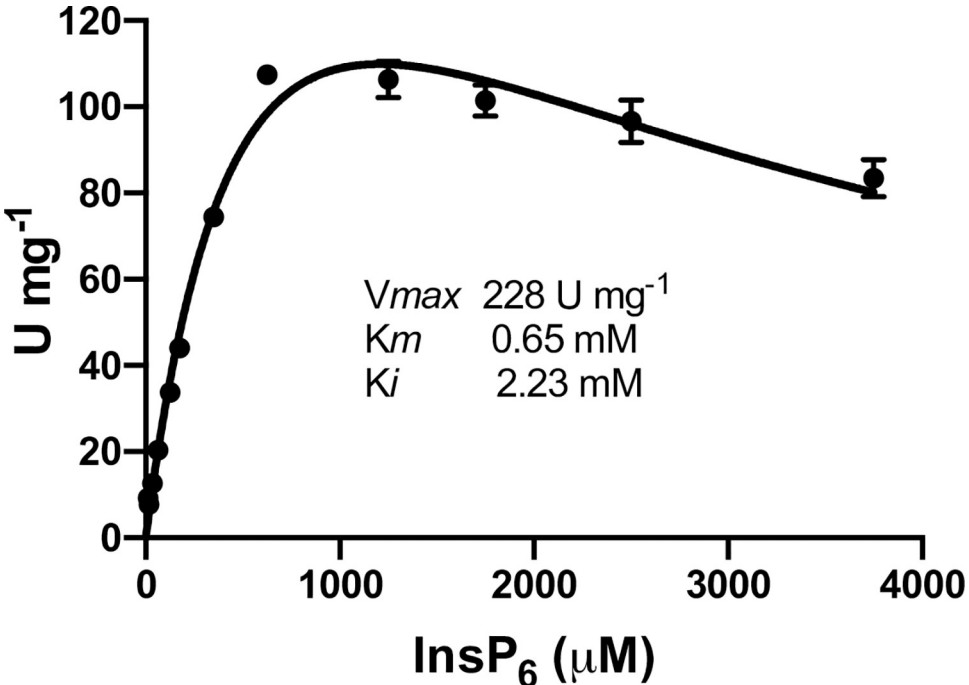

**Fig 7. Kinetics parameters of AC1-2 MINPP activity against phytate.**

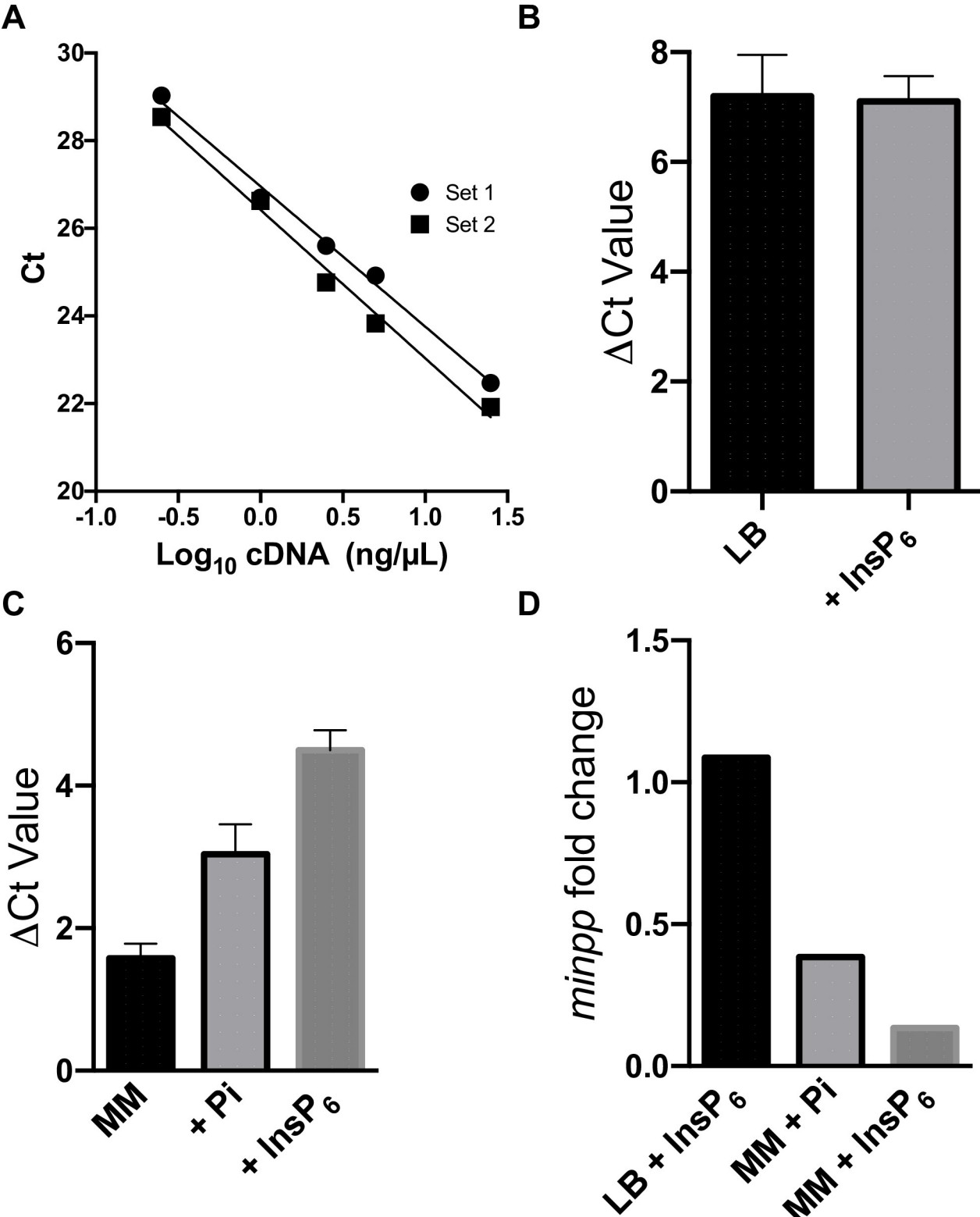

**Fig 8. Validation of the primer sets and quantification of expression of the *ac1-2 MINPP* by qPCR. A**, Log-linearity of amplification with primer sets; **B**, $\Delta$Ct value of the LB and LB + InsP$_6$ environments; **C**, $\Delta$Ct of the MM, MM + Pi and MM + InsP$_6$ environments; **D**, Fold change, calculated using the $2^{-\Delta\Delta Ct}$ method.

phosphate and phytate. Moreover, phytase activity is regulated at both transcriptional and post-translational, levels: AC1-2 down regulates transcription of the *MINPP* gene at levels of substrate approaching K$_i$. In other species, phytase production has been shown to be sensitive to growth phase, being suppressed during the exponential phase in *E. coli* and *Raoultella terrigena*, with resumption of expression upon entering stationary phase [73]. There is still much to be understood about the induction and repression of phytase genes, the expression of which does not appear to be uniformly controlled amongst bacteria [71].

## Conclusions

AC1-2 MINPP, isolated from soil, is a MINPP phytase that displays 5-phytase and D6-phytase activity among other activities. It shows extraordinary long-term stability. The enzyme exhibits many desirable traits that suit its development for use in the animal feed industry.

## Supporting information

**S1 Dataset.**
(XLSX)

**S1 File.**
(PDF)

## Author Contributions

**Conceptualization:** Jonathan D. Todd, Charles A. Brearley.

**Formal analysis:** Hayley Whitfield, Andrew M. Hemmings.

**Funding acquisition:** Charles A. Brearley.

**Investigation:** Gregory D. Rix, Colleen Sprigg, Hayley Whitfield.

**Methodology:** Hayley Whitfield, Andrew M. Hemmings, Charles A. Brearley.

**Supervision:** Jonathan D. Todd, Charles A. Brearley.

**Writing – original draft:** Gregory D. Rix.

**Writing – review & editing:** Andrew M. Hemmings, Charles A. Brearley.

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
