## [Decision Letter · Decision Letter 0]

24 May 2022

PONE-D-22-06931Characterisation of a Soil MINPP Phytase with Remarkable Long-Term Stability and Activity from Acinetobacter sp.PLOS ONE

Dear Dr. Brearley,

Thank you for submitting your manuscript to PLOS ONE. After careful consideration, we feel that it has merit but does not fully meet PLOS ONE’s publication criteria as it currently stands. Therefore, we invite you to submit a revised version of the manuscript that addresses the points raised during the review process. In your revised manuscript please address as fully as possible the highly critical and constructive comments of Reviewer 1.

We look forward to receiving your revised manuscript.

Kind regards,

Israel Silman

Academic Editor

PLOS ONE

Journal Requirements:

“GDR was funded by the Natural Environment Research Council (NERC) PhD studentships (NERC Doctoral Training Programme grant NE/L002582/1) with support from AB Vista.”

“GDR

Natural Environment Research Council (NERC) PhD studentships (NERC

Doctoral Training Programme grant NE/L002582/1)

https://nerc.ukri.org/

Reviewers' comments:

Reviewer's Responses to Questions

**Comments to the Author**

1. Is the manuscript technically sound, and do the data support the conclusions?

Reviewer #1: No

Reviewer #2: Yes

2. Has the statistical analysis been performed appropriately and rigorously? 

Reviewer #1: No

Reviewer #2: No

3. Have the authors made all data underlying the findings in their manuscript fully available?

Reviewer #1: Yes

Reviewer #2: Yes

4. Is the manuscript presented in an intelligible fashion and written in standard English?

Reviewer #1: Yes

Reviewer #2: Yes

5. Review Comments to the Author

Reviewer #1: Phytases are routinely added to farm animal feed to improve phosphate bioavailability from animal feed, and to reduce environmental phosphate contamination. The authors note that research is ongoing to identify and/or engineer phytases with improved catalytic activity, pH activity profile, heat stability, and to lower costs. Into this mix they now add the characteristics of a AC1-2 MINPP, a MINPP-like phytase form a soil Acinetobactor sp, that was recently published by the authors’ laboratory (ref 36). However, there are a number of problems with this work.

1. Figure 1 show the phylogenetic relationship of AC1-2 MINPP to other phytases. This seems unnecessary, because a conceptually similar figure is provided in the author’s previous publication (ref 36). Yes, the current one analyzes more proteins, but no new conclusions are evident.

2. In the absence of real structure, it is understandable that the authors provide a proposed structure obtained by homology modeling based on the BlMINPP template. Lines 221 to 269 discusses this model, leading to predictions about catalytic contributions of a so-called U-loop, and predictions about residue determinants of positional specificity towards individual phosphate groups.

But it is ‘only’ a model (‘proposed’ structural features. . .would have made for a more appropriate header for Figure 4). None of the shortcomings of this modeling have been acknowledged by the authors. In particular, the structure of flexible loops (such as MINPP’s U-loop) is particularly prone to predictive inaccuracy. Moreover, the authors do not model into the active site the natural substrate (inositol hexakisphosphate; IP6), but instead favor a non-metabolized inositol hexasulfate (IS6) substrate analogue (based on a previous paper: ref 11). Furthermore, in their homology modeling template, IS6 exhibits conformational disorder (that is, it binds in multiple orientations). Surely it must be questioned how reliable are these efforts to explain positional specificity using a mobile substrate analogue. To further subtract confidence from the model, the template displays a conformationally active (induced fit) reaction cycle – how reliable, then, is a static model? Another factor that impinges on the reliability of these predictions is not knowing whether it is the 4- or 6-phosphate that is initially hydrolyzed, and yet the authors are willing to attribute six polar contacts between the protein and this (incompletely determined) scissile phosphate.

3. MINPP protein was purified by a two step procedure – a Histrap column followed by gel filtration. Considering the relative simplicity of this protocol, the authors should have provided a silver-stained gel so that purity – and hence the accuracy of the specific activity data - could then be properly assessed.

4. Line 212. Is it really safe to “assume” that the axial 2-phosphate of any inositol phosphate cannot be hydrolyzed by this enzyme, just because it is not removed from IP6? (lines 119-120). Or maybe the 2-phosphate IS removed and the I(1,3,4,5,6)P5 product is so rapidly hydrolyzed it does not accumulate?

5. AC1-2 MINPP is 26% as active against para-nitrophenyl phosphate (PNP) compared to IP6 (does this really qualify as being ‘low’ activity (line 282)? This substrate preference is said to be in agreement with a previous study of avian and plant MINPPs (ref 12,49). I checked and the cited references give 45-48% for plant MINPPS (PNP vs IP6) and only 7% for avian MINPP. I’m not sure how much either result is ‘in agreement’ with the current analysis. That being said, the substrate preferences will also be influenced by the PNP concentration used and its Km value; this important information is not provided by the authors.

6. line 295 and lines 340-341. A little too much is made of the ‘third pH maxima’ of AC1-2 MINPP (Fig 5C), which almost qualifies as an inflection point, rather than a peak, since the activities at pH 3 and 3.5 are barely different. Also, Fig. 5A does not have error bars.

7. Fig. 6. This figure shows enzyme activity over an 800-day time period when the enzyme was exposed to ambient lab temperature. The apparent fall in activity around day 200 with subsequent recovery is a strange observation that gets no mention. In any case, the scientific value of this experiment is questionable since it would be impossible to reproduce. Contemporaneous monitoring of maximum and minimum ambient temperatures, plus gel analysis to look for possible degradation, and a comparison with other MINPP/phytase enzymes, could together have elevated the value of this experiment.

In any case, the authors indicate in the Fig 6 legend that this time course included periods when heatwaves caused ambient temperatures to reach 30 to 35 degrees for several days. The conclusion that during this time MINPP exhibits ‘remarkable long-term stability’ (this phrase is in the title) is not consistent with the data in Fig. 5d, which shows a mere 10 min at 37 degrees causes activity to fall by about 60%.

8. Lines 366-370. Here, the authors make clear that the activity of AC1-2 MINPP (228 U/mg) is below the ‘record highs’ for this enzyme class (1123-2000 U/mg) and only slightly above the more general range of values of other phytases reported in the literature (23-196 U/mg). In that sense, AC1-2 MINPP is not especially notable as a new starting point for engineering higher phytase activity for addition to farm animal feed.

Reviewer #2: The article by Rix and co-workers represents a solid and important piece of work. The article contributes new insights and significantly increases understanding of the potential of phytases. In this case, it concerns MINPP phytase from soil, a phytase n with some very interesting properties.

The necessary experiments have been performed and the manuscript is well structured and well written.

I have only a simple concern with the present manuscript and it deals with the lack of statistical analysis in Figures 5A and 5B. A statical analysis will strengthen the manuscript considerably. And, it should be a simple thing to add.

6. PLOS authors have the option to publish the peer review history of their article (what does this mean?). If published, this will include your full peer review and any attached files.

Reviewer #1: No

Reviewer #2: No

---

## [Author Response · Author response to Decision Letter 0]

30 Jun 2022

We have added a file to the supplementary information

“GDR was funded by the Natural Environment Research Council (NERC) PhD studentships (NERC Doctoral Training Programme grant NE/L002582/1) with support from AB Vista.”

We note that you have provided additional information within the Acknowledgements Section that is not currently declared in your Funding Statement.

We have removed the Acknowledgements Section and the following paragraph describing contributions, as both sets of information are provided in the online submission form. The Funding Statement includes details accompanying a new author who conducted new experiments designed to answer the criticisms of Reviewer 1.

GDR was funded by a Natural Environment Research Council (NERC) PhD studentship (NERC

Doctoral Training Programme grant NE/L002582/1) with support from AB Vista

HW was funded by a Natural Environment Research Council grant (NE/W000350/1)

https://nerc.ukri.org/

Reviewers' comments:

Reviewer's Responses to Questions

Comments to the Author

1. Is the manuscript technically sound, and do the data support the conclusions?

Reviewer #1: No

Reviewer #2: Yes

2. Has the statistical analysis been performed appropriately and rigorously? 

Reviewer #1: No

Reviewer #2: No

3. Have the authors made all data underlying the findings in their manuscript fully available?

Reviewer #1: Yes

Reviewer #2: Yes

4. Is the manuscript presented in an intelligible fashion and written in standard English?

Reviewer #1: Yes

Reviewer #2: Yes

5. Review Comments to the Author

Reviewer #1: Phytases are routinely added to farm animal feed to improve phosphate bioavailability from animal feed, and to reduce environmental phosphate contamination. The authors note that research is ongoing to identify and/or engineer phytases with improved catalytic activity, pH activity profile, heat stability, and to lower costs. Into this mix they now add the characteristics of a AC1-2 MINPP, a MINPP-like phytase form a soil Acinetobactor sp, that was recently published by the authors’ laboratory (ref 36). However, there are a number of problems with this work.

We thank our reviewers for their input. In our address to their considered criticism, we think the manuscript is much improved. The extra experiments that we describe demand the addition of a new author (Hayley Whitfield) and funding details.

1. Figure 1 show the phylogenetic relationship of AC1-2 MINPP to other phytases. This seems unnecessary, because a conceptually similar figure is provided in the author’s previous publication (ref 36). Yes, the current one analyzes more proteins, but no new conclusions are evident.

We have retained the figure, but have moved it to Supplementary Information (Figure S1). The extra information may be useful to some readers.

2. In the absence of real structure, it is understandable that the authors provide a proposed structure obtained by homology modeling based on the BlMINPP template. Lines 221 to 269 discusses this model, leading to predictions about catalytic contributions of a so-called U-loop, and predictions about residue determinants of positional specificity towards individual phosphate groups.

We thank the reviewer for their thoughtful comments on our modelling procedure and positive criticisms of our interpretation of results. These have prompted further experiments, which we think are quite illuminating, and some reappraisal of this manuscript’s findings.

But it is ‘only’ a model (‘proposed’ structural features. . .would have made for a more appropriate header for Figure 4). 

New Figure 4C title: Residues predicted to contribute to the specificity pockets of AC1-2 and other selected histidine phytases.

None of the shortcomings of this modeling have been acknowledged by the authors. In particular, the structure of flexible loops (such as MINPP’s U-loop) is particularly prone to predictive inaccuracy. 

We accept this criticism and have diluted our discussion accordingly. We have also included the following statement.

“However, it is relevant to note that the nature of the homology modelling process, particularly with respect to the prediction of the conformation of large polypeptide loops, leaves room for considerable uncertainty in the conformation of the U-loop in our model of AC1-2.”

Moreover, the authors do not model into the active site the natural substrate (inositol hexakisphosphate; IP6), but instead favor a non-metabolized inositol hexasulfate (IS6) substrate analogue (based on a previous paper: ref 11).

Our modelling was informed by the knowledge that myo-inositol hexakisphosphate is the substrate and that myo-inositol hexakissulfate (InsS6) is commonly reported to act as a competitive inhibitor of phytases and in crystal structures is assumed to mimic the substrate by adopting a pseudo-productive binding mode (Chu et al., 2004; Zeng et al., 2011; Stentz et al., 2014; Acquistapace et al., 2020). As such, we took the minimally invasive step of simply transferring the coordinates of the InsS6 ligand from the homology modelling template structure (PDB 6RXE) to our homology model and replacing the sulfate groups with phosphates. We have further modified the text to reflect this. 

Furthermore, in their homology modeling template, IS6 exhibits conformational disorder (that is, it binds in multiple orientations). Surely it must be questioned how reliable are these efforts to explain positional specificity using a mobile substrate analogue. 

MINPPs, unlike the mineralizing histidine acid phytases, are relatively undiscriminating in their selection of the first phosphate to release from phytate. In our original work, we presumed AC1-2 to be a preferential 6-phytase as no mineralizing enzymes of the histidine phosphatase family have been shown to possess 4- positional specificity. Consequently, we transferred only that conformation of the inhibitor with the 6-sulfate bound in the enzyme A specificity pocket to generate a model for holo-AC1-2. Prompted by this reviewer’s comments we reconsidered the structure of PDB 6RXE and discovered that binding of the 6- or 4-sulfate groups in the A-pocket of BlMINPP has no influence of the residue contacts in any of the specificity pockets except D and E. This is because the sulfate groups of the two binding poses are essentially superimposable in pockets A, B, C and F. In fact, in PDB 6RXE, docking the 4-sulfate group into the A-pocket instead of the 6-sulfate leads to the loss of only two contacts in each of pockets D and E. If InsS6 can be taken to be a true pseudosubstrate inhibitor, then these small differences may form the basis for discrimination between preferential 4- and 6-phytase activity. However, in our modelling we are only attempting to distinguish those interactions in specificity pockets which differ between 4/6-, 5- and 3-phytases and, as such, this difference is unimportant. We have modified the text to explain these differences.

To further subtract confidence from the model, the template displays a conformationally active (induced fit) reaction cycle – how reliable, then, is a static model? 

The fact that the enzyme is predicted to undergo α-domain rotation on substrate binding due to the presence of a U-loop is essentially immaterial to the prediction of the interactions with substrate in the specificity pockets in the productive substrate-bound state. The reliability of our prediction depends only on those interactions involving U-loop residues for which we acknowledge a high degree of uncertainty exists due to the inaccuracies of the modelling process.

Another factor that impinges on the reliability of these predictions is not knowing whether it is the 4- or 6-phosphate that is initially hydrolyzed, and yet the authors are willing to attribute six polar contacts between the protein and this (incompletely determined) scissile phosphate.

PDB 6RXE shows InsS6 to exhibit static disorder in its complex with BlMINPP with both 4- and 6-sulfates bound in the catalytic A-pocket in separate poses. This observation is consistent with the evidence from HPLC that the enzyme displays 4- and/or 6-phytase positional specificity. The 4- and 6-sulfate groups of the two poses are superimposable and indistinguishable. If we presume that InsS6 is a true substrate mimic and adopts a pseudo-productive binding mode, then when bound in the A-specificity pocket the interactions of both the 4- and 6-phosphate groups of phytate with active site residues must also be indistinguishable. Hence, both will be assigned six identical polar contacts.

Nevertheless, we describe additional experiments (Figure 2 with associated text) which employ an enzyme that distinguishes between D-Ins(1,2,3,4,5)P5 and D-Ins(1,2,3,5,6)P5 to show that the InsP5 products of AC1-2 MINPP-catalyzed dephosphorylation of InsP6 include D-Ins(1,2,3,4,5)P5. Consequently, our modelling as a 6-phytase is valid. We do not discount additional 4-phytase activity.

3. MINPP protein was purified by a two step procedure – a Histrap column followed by gel filtration. Considering the relative simplicity of this protocol, the authors should have provided a silver-stained gel so that purity – and hence the accuracy of the specific activity data - could then be properly assessed.

We have now provided gel images in Figure 6 and Fig. S4, albeit not silver stained. 

4. Line 212. Is it really safe to “assume” that the axial 2-phosphate of any inositol phosphate cannot be hydrolyzed by this enzyme, just because it is not removed from IP6? (lines 119-120). Or maybe the 2-phosphate IS removed and the I(1,3,4,5,6)P5 product is so rapidly hydrolyzed it does not accumulate?

We take the reviewer’s point, but find it harder to propose that the absence of a particular InsP5 product can be taken as evidence of its generation. We point out that we should perhaps have stated that there is no precedent for 2-phytases in the MINPP or Histidine Acid Phytase literature. We also note that within phytase classes testing of (lack of) removal of 2-P was explicitly tested for BtMINPP (Stentz et al. 2014 [13] cited in the text). We now make these arguments in the narrative. 

We also describe new experiments designed to identify InsP4 products, discounting Ins(1,3,4,5)P4 and Ins (1,4,5,6)P4 or their enantiomers as products of AC1-2 MINPP action on InsP6 (Figure S2)

It remains the case that the most parsimonious explanation of the absence of Ins(1,3,4,5,6)P5 among products, is that the enzyme does not remove the 2-phosphate of InsP6.

5. AC1-2 MINPP is 26% as active against para-nitrophenyl phosphate (PNP) compared to IP6 (does this really qualify as being ‘low’ activity (line 282)? This substrate preference is said to be in agreement with a previous study of avian and plant MINPPs (ref 12,49). I checked and the cited references give 45-48% for plant MINPPS (PNP vs IP6) and only 7% for avian MINPP. I’m not sure how much either result is ‘in agreement’ with the current analysis. That being said, the substrate preferences will also be influenced by the PNP concentration used and its Km value; this important information is not provided by the authors.

We have modified the description of the results presented in Figure 5 and have provided details of the substrate concentration. We do not feel it necessary in describing a phytase to enquire of the Km for PNP, not least since the enzyme was isolated in a screen for phytase degrading isolates.

6. line 295 and lines 340-341. A little too much is made of the ‘third pH maxima’ of AC1-2 MINPP (Fig 5C), which almost qualifies as an inflection point, rather than a peak, since the activities at pH 3 and 3.5 are barely different. Also, Fig. 5A does not have error bars.

We have modified the text (addition of reference to the literature (11) and citations therein) to describe contemporaneous description of similar pH profiles in other MINPPs. We have added the error bars and statistical test to original 5A (Figure 5A).

7. Fig. 6. This figure shows enzyme activity over an 800-day time period when the enzyme was exposed to ambient lab temperature. The apparent fall in activity around day 200 with subsequent recovery is a strange observation that gets no mention. 

We have added comment to the narrative.

In any case, the scientific value of this experiment is questionable since it would be impossible to reproduce. Contemporaneous monitoring of maximum and minimum ambient temperatures, plus gel analysis to look for possible degradation, and a comparison with other MINPP/phytase enzymes, could together have elevated the value of this experiment.

We wholly disagree, an enzyme retaining activity for greater than 2 years without refrigeration is remarkable. Following the reviewer's suggestion, we have added a poly acrylamide gel of our 3 year-old ‘bench-stored’ enzyme. For this we undertook analysis of protein stored in sucrose, sucrose and BSA, glycerol and glycerol and BSA. The results (Figure S4) show that AC1-2 MINPP has not been degraded during storage for 3 y on the bench. 

In any case, the authors indicate in the Fig 6 legend that this time course included periods when heatwaves caused ambient temperatures to reach 30 to 35 degrees for several days. The conclusion that during this time MINPP exhibits ‘remarkable long-term stability’ (this phrase is in the title) is not consistent with the data in Fig. 5d, which shows a mere 10 min at 37 degrees causes activity to fall by about 60%.

We have undertaken extra experiments with enzyme that has been on the bench for > 3y. These experiments are described in the text and are shown in a new figure (Figure 2). They make it clear that the enzyme has extraordinary stability. We retain the data of original Figure 5D as Figure 6.

8. Lines 366-370. Here, the authors make clear that the activity of AC1-2 MINPP (228 U/mg) is below the ‘record highs’ for this enzyme class (1123-2000 U/mg) and only slightly above the more general range of values of other phytases reported in the literature (23-196 U/mg). In that sense, AC1-2 MINPP is not especially notable as a new starting point for engineering higher phytase activity for addition to farm animal feed.

We disagree, the extreme stability of this enzyme makes it clear that the scaffold could be a good starting point for engineering this class of enzyme, or for searches for related enzymes. We make the point that the commercial position rests on much more than simple specific activity and that current commercial enzymes have been engineered to increase thermostability (in the case of E coli AppA-derivatives, Tm has been increased by more than 12 degrees, probably closer to 20 degrees since first description). We have added this argument to the text with a contemporary reference (72).

Two new references have been added for the discussion of modelling

Chu, H. M. et al. (2004) ‘Structures of Selenomonas ruminantium phytase in complex with persulfated phytate: DSP phytase fold and mechanism for sequential substrate hydrolysis’, Structure. Structure, 12(11), pp. 2015–2024. doi: 10.1016/j.str.2004.08.010.

Zeng, Y. F. et al. (2011) ‘Crystal structures of Bacillus alkaline phytase in complex with divalent metal ions and inositol hexasulfate’, Journal of Molecular Biology. Academic Press, 409(2), pp. 214–224. doi: 10.1016/j.jmb.2011.03.063.

Reviewer #2: The article by Rix and co-workers represents a solid and important piece of work. The article contributes new insights and significantly increases understanding of the potential of phytases. In this case, it concerns MINPP phytase from soil, a phytase with some very interesting properties.

The necessary experiments have been performed and the manuscript is well structured and well written.

I have only a simple concern with the present manuscript and it deals with the lack of statistical analysis in Figures 5A and 5B. A statical analysis will strengthen the manuscript considerably. And, it should be a simple thing to add.

The error bars and statistical analysis has been added to the figure (Figure 5).

---

## [Decision Letter · Decision Letter 1]

12 Jul 2022

Characterisation of a Soil MINPP Phytase with Remarkable Long-Term Stability and Activity from Acinetobacter sp.

PONE-D-22-06931R1

Dear Dr. Brearley,

We’re pleased to inform you that your manuscript has been judged scientifically suitable for publication and will be formally accepted for publication once it meets all outstanding technical requirements.

Kind regards,

Israel Silman

Academic Editor

PLOS ONE

Additional Editor Comments (optional):

Reviewers' comments:

Reviewer's Responses to Questions

**Comments to the Author**

1. If the authors have adequately addressed your comments raised in a previous round of review and you feel that this manuscript is now acceptable for publication, you may indicate that here to bypass the “Comments to the Author” section, enter your conflict of interest statement in the “Confidential to Editor” section, and submit your "Accept" recommendation.

Reviewer #1: All comments have been addressed

2. Is the manuscript technically sound, and do the data support the conclusions?

Reviewer #1: Yes

3. Has the statistical analysis been performed appropriately and rigorously? 

Reviewer #1: Yes

4. Have the authors made all data underlying the findings in their manuscript fully available?

Reviewer #1: Yes

5. Is the manuscript presented in an intelligible fashion and written in standard English?

Reviewer #1: Yes

6. Review Comments to the Author

Reviewer #1: I have reviewed the revised ms and the supporting letter from the authors. The authors have carefully addressed my concerns and the ms should now be accepted.

7. PLOS authors have the option to publish the peer review history of their article (what does this mean?). If published, this will include your full peer review and any attached files.

Reviewer #1: No

---

## [Editor Report · Acceptance letter]

19 Aug 2022

PONE-D-22-06931R1 

Characterisation of a Soil MINPP Phytase with Remarkable Long-Term Stability and Activity from *Acinetobacter* sp. 

Dear Dr. Brearley:

I'm pleased to inform you that your manuscript has been deemed suitable for publication in PLOS ONE. Congratulations! Your manuscript is now with our production department. 

Kind regards, 

on behalf of

Prof. Israel Silman 

Academic Editor

PLOS ONE